# Visual Explanation using Attention Mechanism in Actor-Critic-based Deep Reinforcement Learning

## Abstract

Deep reinforcement learning (DRL) has great potential for acquiring the optimal action in complex environments such as games and robot control. However, it is difficult to analyze the decision-making of the agent, i.e., the reasons it selects the action acquired by learning. In this work, we propose Mask-Attention A3C (Mask A3C) that introduced an attention mechanism into Asynchronous Advantage Actor-Critic (A3C) which is an actor-critic-based DRL method, and can analyze decision making of agent in DRL. A3C consists of a feature extractor that extracts features from an image, a policy branch that outputs the policy, value branch that outputs the state value. In our method, we focus on the policy and value branches and introduce an attention mechanism into them. The attention mechanism applies a mask processing on the feature maps of each branch using mask-attention that expresses the judgment reason for the policy and state value with a heat map. We visualized mask-attention maps for games on the Atari 2600 and found we could easily analyze the reasons behind an agent's decision-making in various game tasks. Furthermore, experimental results showed that the agent's higher performance could be achieved by introducing the attention mechanism.

## 1 Introduction

Reinforcement learning (RL) problems seek optimal actions to maximize cumulative rewards. Unlike supervised learning problems, RL problems collect training data by exploring the environment. Therefore, it is achieving high performance in specific tasks (e.g., controlling autonomous systems (Kober et al., 2013; Gu et al., 2017; Rajeswaran et al., 2018) and video games (Tessler et al., 2017; Justesen et al., 2017; Shao et al., 2019)) that difficult to create training data. In Go, AlphaGo is defeated a professional Go player (Silver et al., 2016). In 2015, the deep Q-network (DQN), a method that combines Q-learning (Watkins & Dayan, 1992) and deep neural network (DNN), achieved a score higher than human players on the Atari 2600 (Mnih et al., 2015). Since the advent of DQN, deep RL (DRL), a method that combines deep learning and RL, has become mainstream, and it is now possible to solve problems featuring a huge number of states, such as images.

In general, deep learning can solve complex tasks by training using a large number of network parameters. However, it is difficult to understand the reasoning behind the decision-making of the trained network because the network parameters used to make the decision are enormous. This problem occurs in DRL as well. The reason for judging the acquired action is unclear, since agents collect training data by searching the environment and the calculation inside the network is complicated. Therefore, in order to prove that the trained network is sufficiently reliable, it is important to analyze the reason for the judgment of the action that it outputs.

One approach to interpreting the decision-making of a network, visual explanation, has been studied in the field of computer vision (Zhou et al., 2016; Selvaraju et al., 2017; Fukui et al., 2019). Visual explanations analyze the factors of the network output by using an attention map that highlights the important regions in an input image. Visual explanation methods have also been applied to DRL models to help with understanding the decision-making of an agent (Sorokin et al., 2015; Greydanus et al., 2018). These methods can be categorized into two approaches: bottom-up and top-down. Bottom-up visual explanations compute attention maps by using the gradient informa-

tion of a network. Because the bottom-up approach does not need to re-train a network, it can be applied to any trained network and is commonly used in computer vision and DRL. The attention maps obtained by bottom-up approach are based on the input data and response values calculated from each layers. The bottom-up approach highlights local textural context. Top-down visual explanations generates attention maps by using the response values in a network. Unlike the bottom-up approach, the attention maps of the top-down approach are output for current network output.

In this paper, we propose Mask-Attention A3C (Mask A3C) which introduce an attention mechanism into Asynchronous Advantage Actor-Critic (A3C) of an actor-critic-based DRL method. Mask A3C calculates an mask-attention that an attention map of policy and state value, and then a visual explanation for these values is achieved by visualizing the acquired mask-attention. Our method also learns the policy and state value while considering mask-attention by implementing the attention mechanism, thereby improving the performance of the agent.

**Contributions**    The main contributions of this paper are as follows.

- We propose a top-down visual explanation method that implements an attention mechanism in the DRL model. In the proposed method, mask-attention, which is an attention map for the outputs, can be obtained simply by forward pass.

- In the proposed method, the decision-making of the agent after learning can be analyzed by visualizing the acquired mask-attention. We conducted an experiment with games on the Atari 2600 and analyzed which information influences the agent's decision-making.

- By implementing the attention mechanism in the policy branch and value branch of the actor-critic method, a different mask-attention can be obtained depending on the policy and state value. In this way, it is possible to analyze an agent's decision-making from the two viewpoints of policy value and state value.

- The proposed method outputs the control value of the agent while considering mask-attention by implementing the attention mechanism. Therefore, the performance of the agent can be improved by emphasizing the area related to the control value.

## 2    RELATED WORKS

### 2.1    DEEP REINFORCEMENT LEARNING

The deep Q-network (DQN) (Mnih et al., 2015), which is a typical method of DRL, expresses the action value function $Q(a|s; \theta)$ by using a neural network and acquires the optimum action by training the network parameters $\theta$. DRL methods that learns the optimal action by a value function such as DQN is called value-based DRL, and have been studied extensively (Van Hasselt et al., 2016; Wang et al., 2016; Bellemare et al., 2017; Hessel et al., 2018). There is also a policy-based DRL that directly learns the policy by expressing the policy $\pi(a|s; \theta)$ with a neural network (Lillicrap et al., 2016; Schulman et al., 2015; 2017; Haarnoja et al., 2018). The actor-critic method (Konda & Tsitsiklis, 2000), which is a policy-based method, consists of an actor that output the policy $\pi(a|s; \theta)$ and a critic that output the state value $V(s; \theta)$. Here, the state value $V(s; \theta)$ numerically expresses how the current state $s$ contributes to the reward. The actor selects and performs an action according to a policy $\pi(a|s; \theta)$ that is a probability distribution from a state $s$ to an action $a$. The critic estimate the state value $V(s; \theta)$ as the evaluation value of the policy $\pi(a|s; \theta)$ that is output by the actor. To update the network parameters in the actor-critic method, the actor parameter update by the policy gradient method and the critic parameter update by the TD error are performed in parallel.

Other approaches include distributed DRL, which improves learning efficiency by constructing multiple environments and agents (Nair et al., 2015; Jaderberg et al., 2017; Kapturowski et al., 2019). Asynchronous Advantage Actor-Critic (A3C) (Mnih et al., 2016) is a distributed DRL method that utilizes the actor-critic method. A3C introduces Asynchronous that is an asynchronous parameter update in distributed learning and Advantage that learns while considering rewards several steps ahead. Experiments with the Atari 2600 showed that A3C could achieved a high score in a short training time by executing the generation of experiences used for learning in parallel.

In this study, we acquire mask-attention, attention maps for policy and state value, by implementing an attention mechanism in A3C. By visualizing mask-attention at inference, the decision-making of the agent acquired by learning is analyzed from the visual explanation of the policy and state value.

## 2.2 VISUAL EXPLANATIONS

**Visual Explanations in Image Recognition**

In the field of image recognition, several methods utilizing an attention map have been proposed for analyzing the reason for judgments on the inference result of the network. Attention map is a map that visualized the network gazed area at the time of inference as a heat map. Zhou et al. (2016) proposed a class activation mapping (CAM), which acquires the attention map of a specific class from the response value of the convolutional layer and the weight of the fully connected layer. However, the recognition performance of CAM deteriorates because it is necessary to change a part of the network structure, such as by introducing global average pooling (GAP) between the convolution and fully connected layers. For that problem, Selvaraju et al. (2017) proposed gradient-weighted CAM (Grad-CAM), which acquires an attention map by using the response value of the convolutional layer during forward pass and the gradient during back-propagation. Grad-CAM avoids the problem that the recognition performance deteriorates by generating an attention map from the gradient information. In image recognition, the recognition accuracy is known to improve by using an attention map during learning. Fukui et al. (2019) proposed an attention branch network (ABN) that applies the attention map to the attention mechanism. This method provides a visual explanation of the reason for judgment by the attention map simultaneously improves the recognition accuracy.

**Visual Explanations in Deep Reinforcement Learning**

In DRL, several works have examined how to analyze the reason for judging an action that output of the network. Sorokin et al. (2015) proposed the deep attention recurrent Q-network (DARQN). They implements an attention mechanism in DQN, a representative value-based method. Manchin et al. (2019) introduces self-attention to a policy-based DRL method, in order to improve the score along with policy analysis. This method analyzes the agent's decision making using an attention map for policy. On the other hand, our method can improve the interpretability of the agent's decision making in the actor-critic based DRL method by simultaneously acquiring different attention maps for policies and state values.

Greydanus et al. (2018) has acquired a saliency map in A3C by calculating a perturbation image utilizing an applied Gaussian filter from the gradient at during back-propagation. This method takes a bottom-up approach is similar to Grad-CAM, and therefore it is necessary to perform back-propagation to acquire the saliency map. In contrast, our method takes a top-down approach that implement an attention mechanism in the network structure of A3C.

Zhang et al. (2018) proposed attention guided imitation learning (AGIL), which guides the focus area of a network on the basis of human gaze information. They trained a model to replicate human attention with supervised gaze heat maps. The input state was then augmented with this additional information. This style of attention fundamentally differs from that used in our work as it incorporates hand crafted features as input.

The most similar work with us is conducted by Mott et al. (2019). They proposed a model for acquiring two attention ("what" and "where") by using query-based attention in an actor-critic based DRL method. This method requires an attention query to be generated, which requires significant changes to the network architecture (e.g., Keys, Values). In contrast, our method has a simple structure in which an attention mechanism is introduced in the policy and value branches and does not require significant changes to the network architecture. Also, they obtained different attentions related to "what" and "where" by generating an attention query. In contrast, our method improves interpretability by acquiring different attentions toward policy and state value, which are the outputs of actor-critic-based DRL methods.

The above visual explanations for DRL generates attention maps from the low-level feature maps extracted from early or middle convolutional layers, or output for action (e.g., policy output or Q value output on DQN models). In DRL, a *strategy* is an important clue to solve given task and environment. From the viewpoint of the strategy, the state value of actor-critic based model plays a

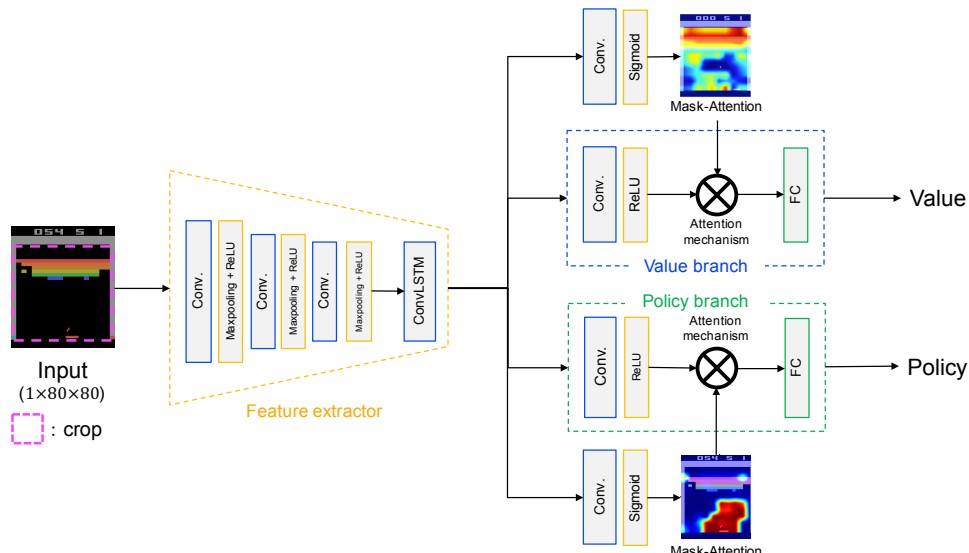

Figure 1: **Detailed network structure of Mask-Attention A3C.**

crucial role because the state value is the expected value from current to future states and affects for the future action selections. However, the existing methods outputs attention maps with respect to the instantaneous action selection. Our method based on an actor-critic outputs two attention maps from both policy and value branches. Considering both attention maps, we can understand the basis of an agent's decision-making in more detail.

## 3 MASK-ATTENTION A3C

We propose Mask-Attention A3C (Mask A3C), that introduces an attention mechanism into A3C, which is an actor-critic-based distributed DRL method. In Mask A3C, by implementing an attention mechanism for the policy branch and value branch, we can acquire an attention map that expresses the focus area of the network for the output of each branch. In our study, this attention map is called mask-attention. By visualizing the mask-attention of each branch, we obtain a visual explanation of the reason for the judgment on the policy and state value. In addition, our method learns while considering the mask-attention by implementing the attention mechanism, which improves the performance of the agent.

### 3.1 OVERVIEW OF MASK A3C STRUCTURE

Figure 1 shows the network structure of the proposed Mask A3C. It consists of a feature extractor, policy branch, value branch, and attention mechanism. The feature extractor calculates the feature map for the input image using the convolutional layer, the policy branch outputs policies, and the value branch outputs the state value function. In A3C, temporal information can be considered by utilizing LSTM, which greatly improves the performance of the agent. However, LSTM cannot consider the spatial information for the input image, so if it is used in Mask A3C, mask-attention cannot be calculated. We therefore employ convolutional LSTM (ConvLSTM) (Xingjian et al., 2015), which can considers spatiotemporal information. Mask-attention is calculated by applying the sigmoid function and one convolutional layer of $1 \times 1 \times$ # of channels to the feature map calculated by the feature extractor.

### 3.2 ATTENTION MECHANISM

Mask A3C implements an attention mechanism in the policy branch and value branch so that the policy and state value function are learned in consideration of the acquired mask-attention. The attention mechanism performs mask processing to the feature map of the middle layer in each branch

using mask-attention. With this mask processing, the area that contributes to the optimum action and state value can be emphasized. This mask processing using mask-attention for the feature map is calculated as

$$F'(\mathbf{s}_t) = F(\mathbf{s}_t) \cdot M(\mathbf{s}_t),  \tag{1}$$

where, $\mathbf{s}_t$ is the state, $F(\mathbf{s}_t)$ is the feature map of the middle layer in each branch, $M(\mathbf{s}_t)$ is mask-attention, and $F'(\mathbf{s}_t)$ is the feature map after mask processing.

## 4 EXPERIMENTS

To evaluate the effectiveness of Mask A3C, we conducted an experiment using the game task of OpenAI gym (Brockman et al., 2016). Three games were used: "Breakout", "Ms.Pac-Man", and "Space Invaders". The comparison methods were A3C, Policy Mask A3C , Value Mask A3C, and Mask A3C, for a total of four patterns. Policy Mask A3C and Value Mask A3C refer to a Mask A3C in which the attention mechanism implemented in one side of branch (i.e., policy branch or value branch). The learning conditions were 35 for the number of workers, 0.0001 for the learning coefficient, and 0.99 for the discount rate. The termination condition of learning was when the global steps reached $1.0 \times 10^8$. The termination condition of an episode was the end of one play in the game and the case where the number of steps reached $1.0 \times 10^4$. We used the following three evaluation methods.

- Visual explanations using mask-attention
- Score comparison on the Atari 2600
- Score comparison by inverting the gaze area of mask-attention

### 4.1 IMPLEMENTATION DETAILS

The input was a grayscale image of the game screen and the output was the action in each game. The image used as input was resized to $80 \times 80$. The output dimension of feature extractor was 32-dimensional for the first two convolutional layers and 64-dimensional for the remaining one convolutional layer. For the hidden state in ConvLSTM, the output dimension was 64-dimensional. The policy branch consisted of one convolutional layer, one fully connected layer, and a softmax function. The output dimension of the convolution layer were 32 dimensional and the number of output units of the fully connected layer was the number of actions in each game. The value branch consisted of one convolutional layer with 32-dimensional output dimension and one fully connected layer with the number of output units of 1. The network structure of A3C in this experiments was the structure excluded the attention mechanism from Mask A3C.

### 4.2 VISUAL EXPLANATIONS USING MASK-ATTENTION

Figures 2 and 3 show visualization examples of mask-attention in Atari 2600. (Examples of other frames and other environment are available in Appendix A.) Hereafter, we discuss the obtained mask-attentions shown in Figs. 2 and 3.

**Breakout** Breakout is a game in which the player hits the ball with a paddle to destroy various block. The actions of the agent that is the paddle are "Noop", "Left", and "Right". As shown in figure 2(a), the agent in Frame 1 attended to the traveling direction of the ball. In Frame 2, the agent attended to the right side of the paddle, and the action was "Right". Frame 3 after the ball was hit back shows no gaze area. From these results, we can see that the agent controlled the paddle according to the traveling direction of the ball. In figure 3(a), the agent of Point 1 was attending to the entire block, and we can see from the graph that the state value increased with the destruction of the blocks. In Point 2, the ball reached the top of the block and destroyed many blocks. At that time, the gaze area of mask-attention was significantly reduced according to the blocks. After Point 2, the state value decreased according to the number of blocks that were destroyed. These results demonstrate that the agent recognized the blocks as the score source.

**Ms.Pac-Man** Ms.Pac-Man is a game in which the player collects scattered cookies while avoiding enemies. The actions of the agent that is the Pac-Man are "Noop","Up","Down","Left","Right","Up

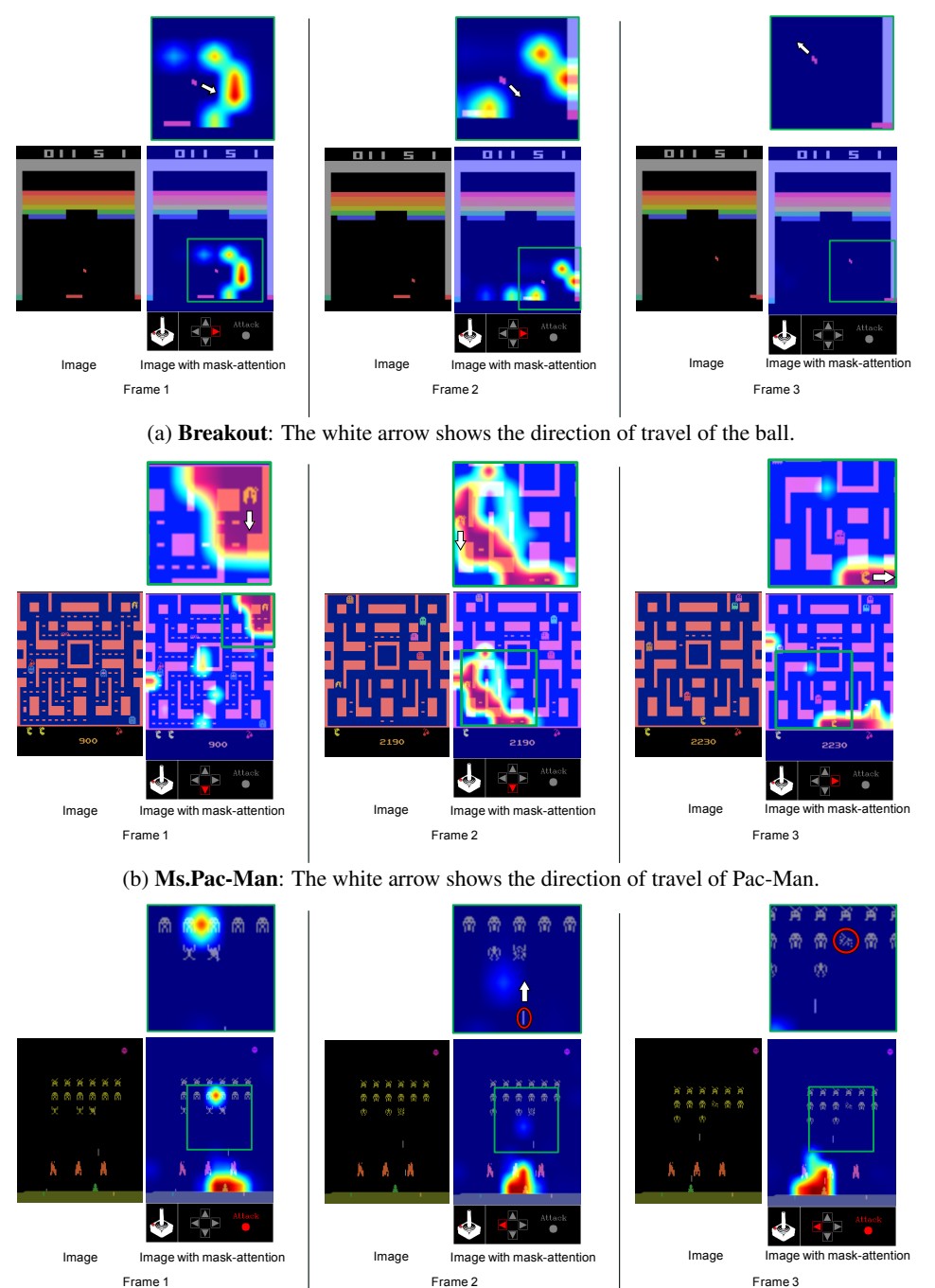

(a) **Breakout**: The white arrow shows the direction of travel of the ball.

(b) **Ms.Pac-Man**: The white arrow shows the direction of travel of Pac-Man.

(c) **Space Invaders**: The red circles in Frame 2 show the beam that is the attack of agent in Frame 1, and the red circles in Frame 3 show the destroyed invaders. The white arrow shows the direction of travel of the beam that is the attack of the agent.

Figure 2: **Visualization example of mask-attention in policy**: The controller in "Image with mask-attention" is action that is output by the DRL model. The green broken line in the State value shows the transition to the next stage.

+ Left","Up + Right","Down + Left", and "Down + Right". In figure 2(b), the agent of Frame 1 was attending around Pac-Man. The agent of Frame 2 was attending to the cookies remaining on the screen. In Frame 3, Pac-Man moved to the gazed area of Frame 2 and acquired the cookie. These

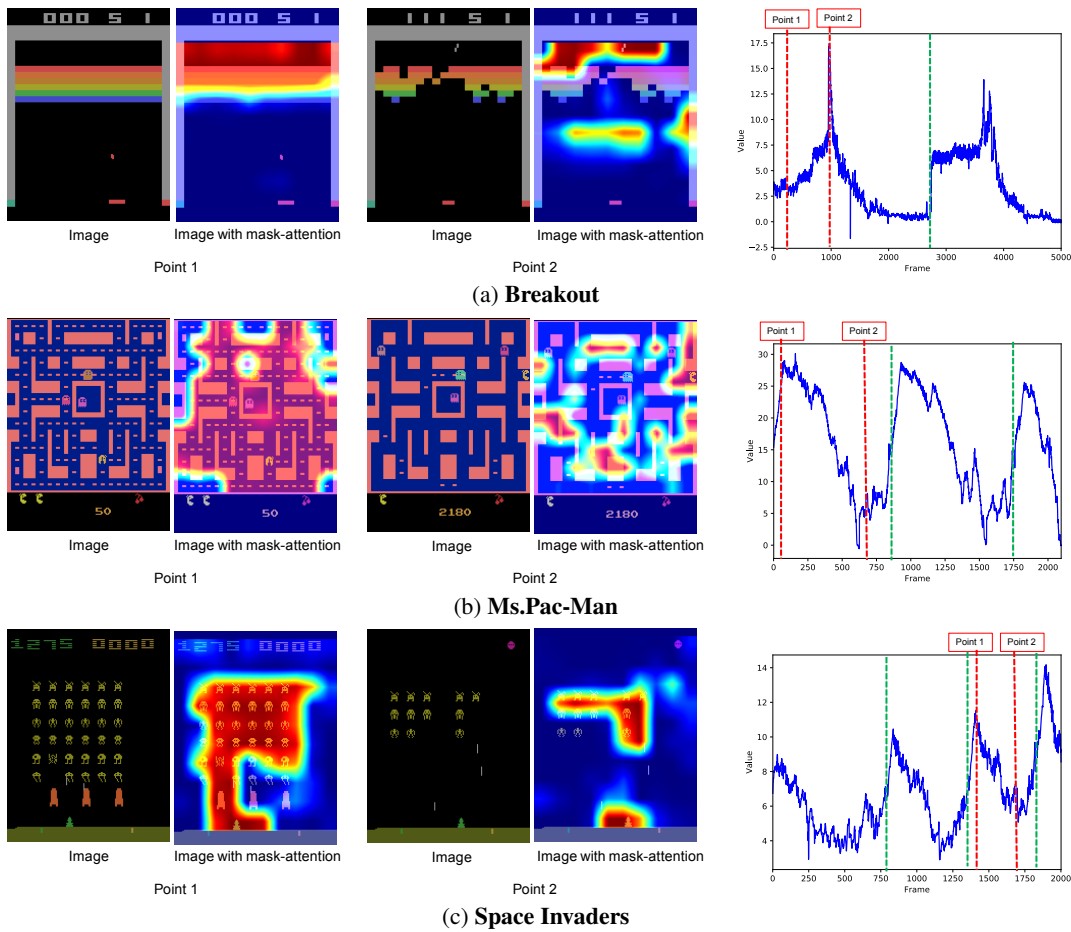

Figure 3: **Visualization example of mask-attention in state value**: The green broken line in graph shows the transition to the next stage.

results demonstrate that the agent controlled Pac-Man toward the cookies while simultaneously attending to the surroundings of Pac-Man. In figure 3(b), Point 1, which was the beginning of the game, was attending at the entire screen. In contrast, Point 2 is reduced the gaze area according to the decreases of cookies. Also, from Point 1 to Point 2, the state value decreased as the cookies on the screen decreased. These results demonstrate that the agent recognized the cookies as the score source.

**Space Invaders**    Space Invaders is a shooting game in which the player repels the enemy invaders. The actions of the agent that is the cannon are "Noop","Left","Right","Attack","Left + Attack", and "Right + Attack". In figure 2(c), the agent in Frame 1 is attended to the invaders and the action was "Attack". In Frame 2, we can see that the beam in Frame 1 was heading toward the invader, and in Frame 3, the beam was repelling the invaders that were attended at in Frame 1. In all frames, the agent attended around itself while avoiding the defensive walls. From these results, we can see that the agent repelled the invaders while simultaneously avoiding the defensive wall. In figure 3(c), the agent in Point 1 was attending to the all of the invaders and the agent in Point 2 was shrinking the gaze area according to the decreasing number of invaders. In addition, from Point 1 to Point 2, the state value is decreased according to the number of invaders. These results demonstrate that the agent recognized the invaders as the score source.

**Discussion**    We obtained different mask-attentions in the policy and state value by implementing an attention mechanism for each branch. The policy represents the probability distribution of the action selection in the current state. Therefore, the mask-attention of the policy indicates the area

Table 1: **Max and mean scores over 100 episodes on Atari 2600**: Scores of models that had the highest average score among five trials in each method are shown.

| Att. mechanism | | Breakout | | Ms.Pac-Man | | Space Invaders | | Beamrider | | Fishing Derby | | Seaquest | |
|---|---|---|---|---|---|---|---|---|---|---|---|---|---|
| Policy | Value | max | mean | max | mean | max | mean | max | mean | max | mean | max | mean |
| | | **864** | **662.0** | 5380 | 4573.3 | 19505 | 18531.8 | 34748 | 28341.1 | 41 | 32.1 | 2760 | 2728.2 |
| ✓ | | **864** | 595.8 | 6330 | 4833.8 | **19860** | 19102.8 | 32604 | **28495.3** | 41 | **37.5** | 2820 | 2784.0 |
| | ✓ | **864** | 606.9 | 4830 | 4044.5 | 19675 | 18537.8 | **35108** | 28205.7 | **43** | 36.1 | 2820 | 2786.4 |
| ✓ | ✓ | **864** | 640.0 | **6610** | **5314.1** | 19810 | **19212.5** | 34448 | 27671.1 | 41 | 34.3 | **17150** | **6701.9** |

that contributes to the action of the agent. The state value represents the expected value of the return in the current state. Here, return is the sum of rewards during one episode. Therefore, the mask-attention of the state value indicates the area that represents the property of the game.

## 4.3 SCORE COMPARISON ON THE ATARI 2600

Table 1 shows the max and mean scores over 100 episodes on the Atari 2600. As we can see, the mean score in Breakout was lower Policy Mask A3C, Value Mask A3C, and Mask A3C than the A3C. In contrast, the max score in Breakout was 864 for all methods. This score is the best score that can be obtained in Breakout. Breakout is a simple game with no external factors: it simply consists of the player hitting the ball back with a paddle. Therefore, we are consider that A3C and Mask A3C got the same score. In Ms.Pac-Man, the max and mean scores of Policy Mask A3C and Mask A3C improved compared to those of A3C. The control of agent in Ms.Pac-Man is complex because it needs to select an action while considering external factors(e.g., the enemy). Policy Mask A3C and Mask A3C, which implements an attention mechanism on the policy branch, can emphasize the areas that contribute to the action (e.g., cookies and enemies), which is why Policy Mask A3C and Mask A3C obtained a higher score than A3C. In Space Invaders, Policy Mask A3C and Mask A3C improved the max and mean scores compared to A3C. Also, the max and mean scores of Value Mask A3C is almost the same as the A3C. The agent in Space Invaders needs to select actions in consideration of external factors (e.g, enemies), the same as in Ms.Pac-Man. Policy Mask A3C and Mask A3C, which implements an attention mechanism in the policy branch, can emphasize the areas that contribute to the action (e.g, defensive walls and invaders), which is why Policy Mask A3C and Mask A3C obtained a higher score than A3C. There is no significant difference between the scores of Beamrider. In Beamrider, two kinds of enemies are exists from the an agent's point of view. One is an enemy that the agent should defeat, and the other is that the agent should avoid collisions. These enemies look like similar appearance. Because our attention-mask highlighting the enemies and an attention mechanism are insufficient to distinct these enemies, these do not contribute the score improvement. In Fishing Derby, Policy Mask A3C, Value Mask A3C, and Mask A3C improved the mean scores compared to A3C. In Fishing Derby, there are many fish that are score source, and getting the closest fish from the player is the fastest way to earn score. Policy Mask A3C, Value Mask A3C, and Mask A3C, which implements an attention mechanism, can emphasize the fish closest to the player. This is the reason why the Mask-attention methods obtained a higher score than that of A3C. In Seaquest, Mask A3C improves both the maximum and average scores compared to the other methods. This result is because only the Mask A3C agent could learn the action of replenishing oxygen. Seaquest is an oxygen gauge at the bottom of the screen, and the game ends when the oxygen is gone. Mask A3C, which implements an attention mechanism in policy and value branch, can emphasize the oxygen gauge, which is why Mask A3C obtained a higher score than other methods.

## 4.4 COMPARISON OF SCORES BY INVERTING GAZE AREA IN MASK-ATTENTION

In visually explaining the decision making of an agent using Mask A3C, we want to verify whether mask-attention represents an effective gaze area for the optimum action. In case that the obtained game score by inverting mask-attention does not change, the mask-attention does not contribute for acquired agent's action. In contrast, if the game score significantly decreases, the mask-attention largely contributes for the actions acquiring the game score. For this verification method, we create a map in which the gaze area of the mask-attention in the policy branch is inverted and then calculate the score on Atari 2600 when the created map is used for the attention mechanism. We determine whether mask-attention is effective for the visual explanation of an action by comparing the scores

Table 2: **Score comparison by inverting the gaze area of mask-attention**: Normal and inverse are the scores when the gaze area is not inverted and is inverted, respectively. Random is the score when the action is randomly selected. Max / mean = maximum and average scores over 100 episodes.

| Att. mechanism | | | Breakout | | Ms.Pac-Man | | Space Invaders | | Beamrider | | Fishing Derby | | Seaquest | |
| Policy | Value | | max | mean | max | mean | max | mean | max | mean | max | mean | max | mean |
|---|---|---|---|---|---|---|---|---|---|---|---|---|---|---|
| ✓ | | normal | 864 | 595.8 | 6630 | 4833.8 | 19860 | 19102.8 | 32604 | 28495.3 | 41 | 37.5 | 2820 | 2784.0 |
| | | inverse | 4 | 2.2 | 290 | 268.9 | 805 | 306.9 | 4996 | 1554.2 | -49 | -75.7 | 280 | 158.2 |
| ✓ | ✓ | normal | 864 | 640.0 | 6610 | 5314.1 | 19810 | 19212.5 | 34448 | 27671.1 | 41 | 34.3 | 17150 | 6701.9 |
| | | inverse | 5 | 1.8 | 410 | 194.4 | 915 | 420.2 | 6380 | 2063.9 | -49 | -74.7 | 420 | 280.6 |
| random | | | 5 | 1.2 | 1080 | 247.8 | 460 | 142.1 | 852 | 356.5 | -85 | -93.1 | 300 | 82.8 |

Table 3: **Decrease rate of score due to inverse of gaze area in mask-attention** (%): Max / mean = maximum and average scores over 100 episodes.

| Att. mechanism | | Breakout | | Ms.Pac-Man | | Space Invaders | | Beamrider | | Fishing Derby | | Seaquest | |
| Policy | Value | max | mean | max | mean | max | mean | max | mean | max | mean | max | mean |
|---|---|---|---|---|---|---|---|---|---|---|---|---|---|
| ✓ | | 99.53 | 99.63 | 95.41 | 94.43 | 95.94 | 98.39 | 84.67 | 94.54 | 98.90 | 99.12 | 90.07 | 94.31 |
| ✓ | ✓ | 99.42 | 99.71 | 93.79 | 96.34 | 95.38 | 97.80 | 81.47 | 92.54 | 98.90 | 99.09 | 97.55 | 95.81 |

when the gaze area is inverted and when it is not inverted. The map in which the gaze area of the mask-attention is inverted is created by

$$M_{\text{inverse}}(\mathbf{s}_t) = 1 - M(\mathbf{s}_t), \tag{2}$$

where, $\mathbf{s}_t$ is the state (grayscale image in the experiment), $M(\mathbf{s}_t)$ is mask-attention, and $M_{\text{inverse}}(\mathbf{s}_t)$ is the map after inverting the gaze area of mask-attention.

Table 2 shows the score comparison by inverting the gaze area of mask-attention. (Also, the table 3 shows for the decrease ratio from the normal score to the inverse score.)

As shown in table 2, the inverse score was significantly lower than the normal score in all games. In addition, the inverse score in Breakout was equivalent to a random score, and the inverse score of Mask A3C in Ms.Pac-Man was 53.4 lower than random. In contrast, we can see that the inverse score in Space Invaders, Beamrider, Fishing Derby and Seaquest was higher than a random score. However, from Table 3, we can see that the decrease ratio in mean of Space Invaders, Beamrider, Fishing Derby, and Seaquest was more than 90%, as with the other games. Therefore, we conclude that the gaze area of mask-attention in the policy branch can represents a useful area for action to obtain a high score.

## 5 CONCLUSION

In this paper, we proposed Mask-Attention A3C (Mask A3C) that introduces an attention mechanism into Asynchronous Advantage Actor-Critic (A3C). In Mask A3C, we acquired a mask-attention that expresses the important area for the policy and state value by implementing an attention mechanism in the policy and value branches. This enables a visual explanation of the judgment reason in the decision-making of the agent, from the two viewpoints of policy and state value, by visualizing mask-attention. We also emphasize which areas contribute to the optimal action and state value by implementing an attention mechanism, and improves the performance of the agent.

Experiments with the Atari 2600 confirmed the acquisition of different mask-attentions in the policy and value branches. The results demonstrate that the the mask-attention of the policy branch indicates the area that contributes to action while that of the value branch indicates the area that expresses the property of the game. We provided a useful analysis for the decision-making of agents in game tasks from the two viewpoints of policy and state value by visualizing these mask-attentions. A comparison of game scores showed that the score improved when the attention mechanism was implemented in the policy branch. However, our experiments were conducted with game tasks that are easy to analyze visually. Our future work will entail the visual analysis of agents using mask-attention is a future work for complex tasks (e.g., robot control and autonomous driving).

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

## A    APPENDIX: ADDITIONAL MASK-ATTENTION EXAMPLES

**Mask-attentions in policy**

Figures 4, 5, 6, 7, 8, and 9 show mask-attentions in policy of each environment.

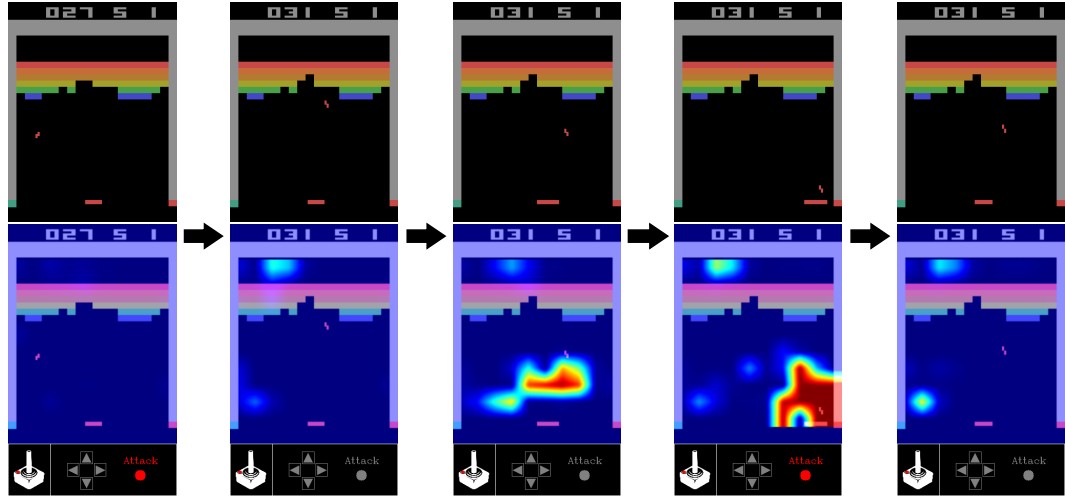

Figure 4: **Mask-attention in policy of Breakout.**

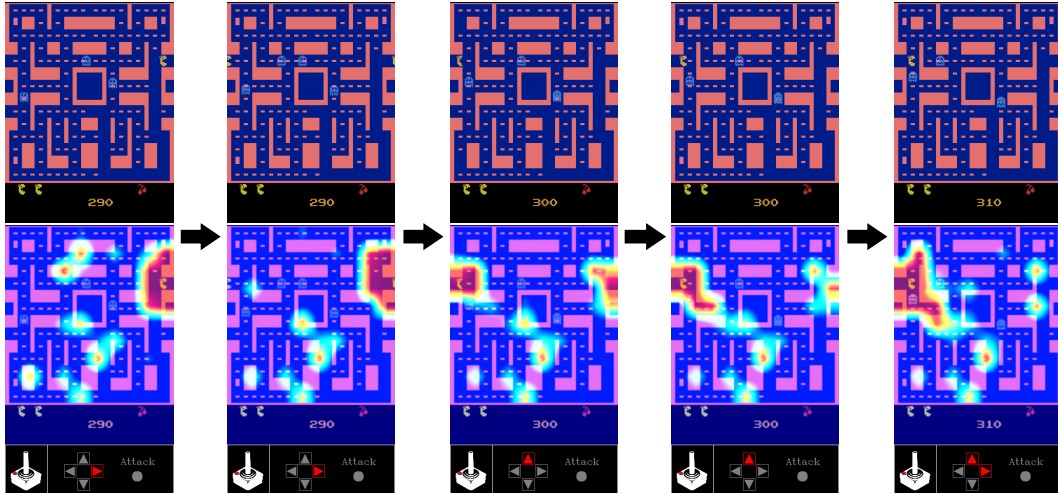

Figure 5: **Mask-attention in policy of Ms.Pac-Man.**

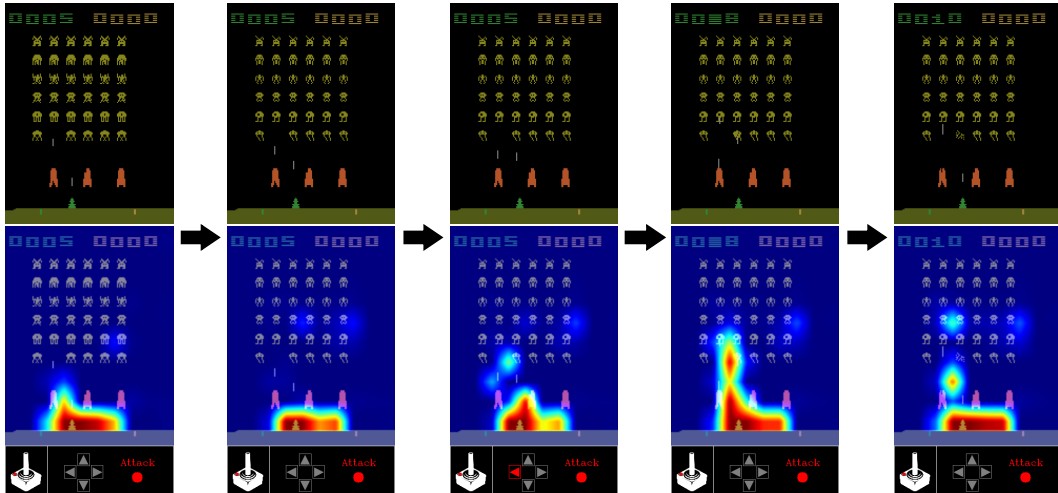

Figure 6: **Mask-attention in policy of Space Invaders.**

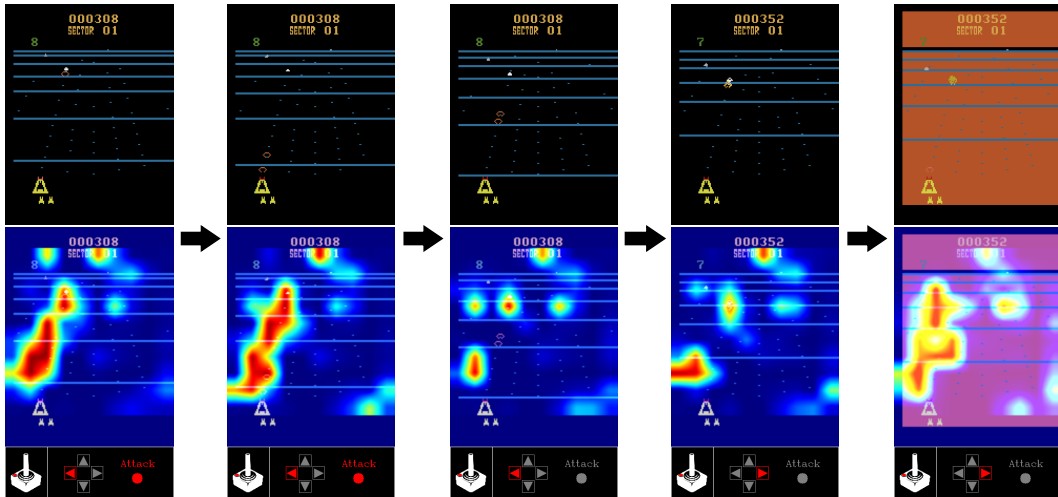

Figure 7: **Mask-attention in policy of Beamrider.**

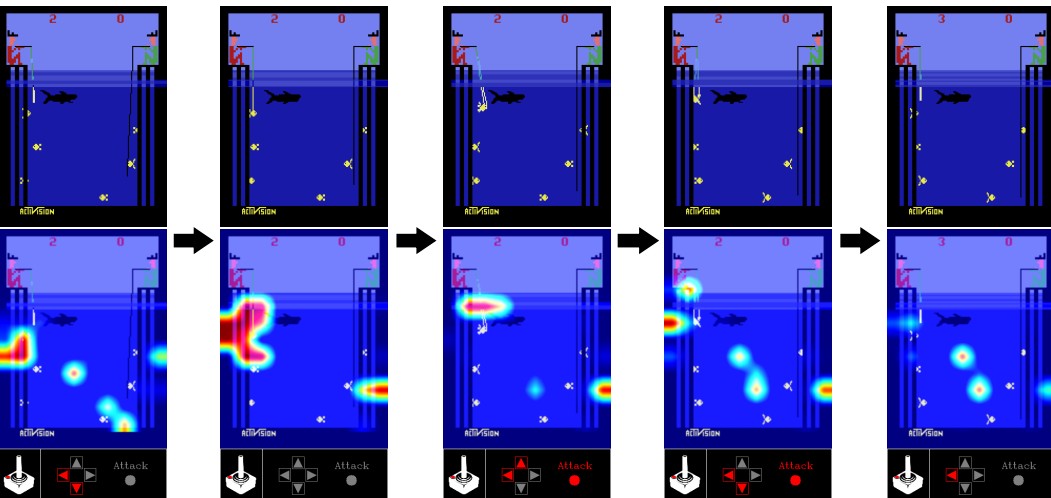

Figure 8: **Mask-attention in policy of Fishing Derby.**

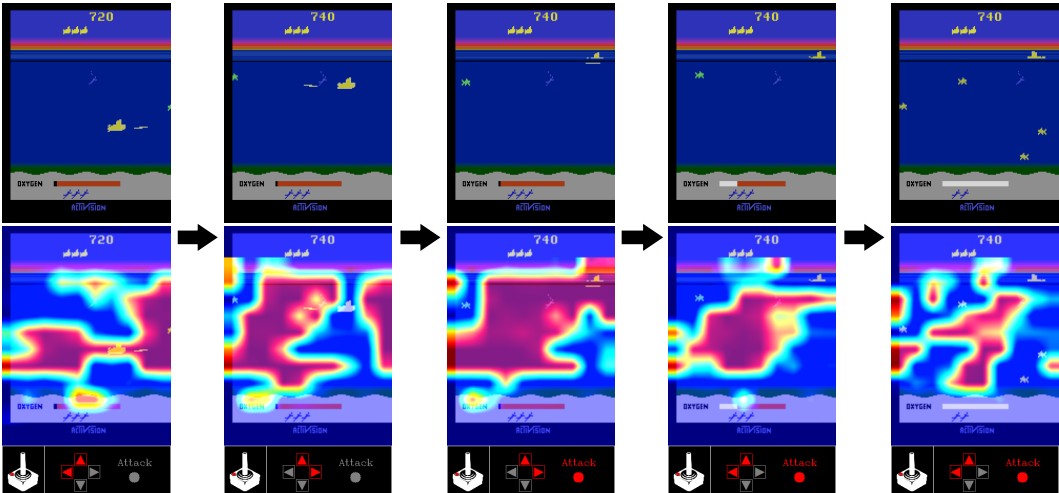

Figure 9: **Mask-attention in policy of Seaquest.**

**Mask-attentions in state value**

Figures 10, 11, and 12 show mask-attentions in state value.

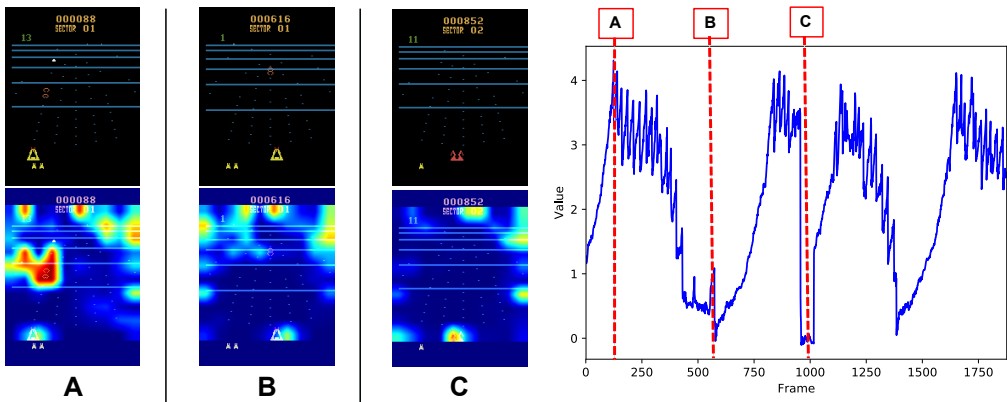

Figure 10: **Mask-attention in value of Beamrider.**

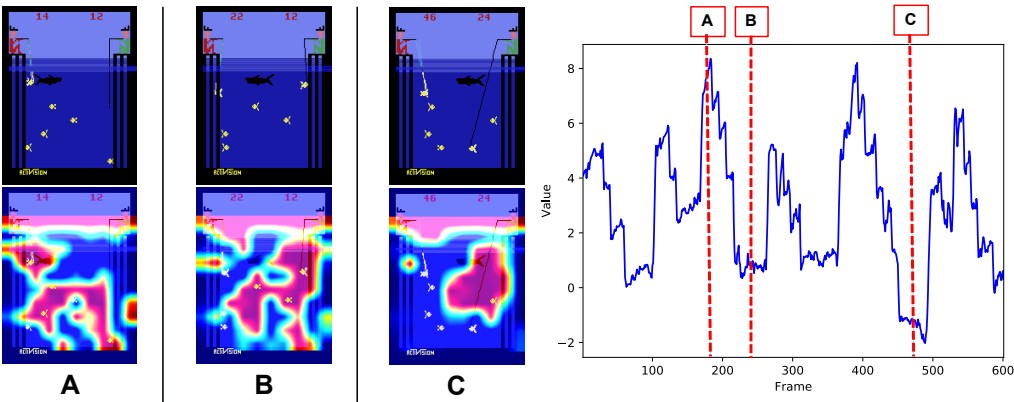

Figure 11: **Mask-attention in value of Fishing Derby.**

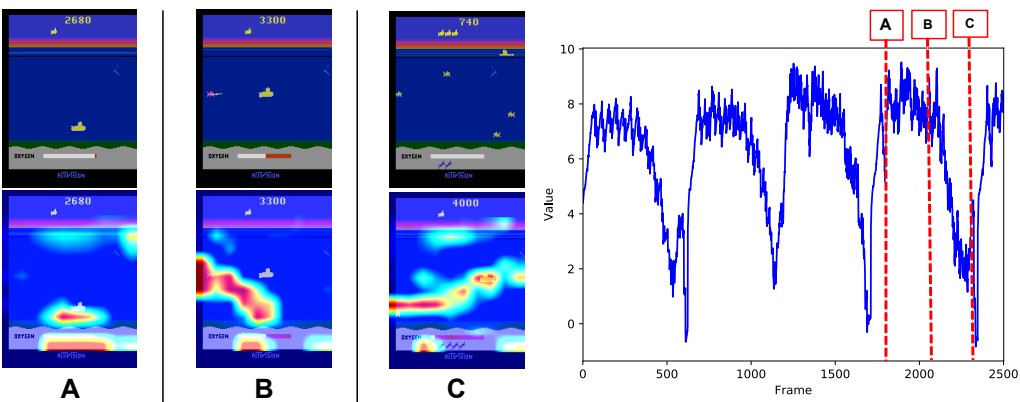

Figure 12: **Mask-attention in value of Seaquest.**

**Transitions of Mask-attentions in policy and state value**

Figures 13, 14, 15, 16, 17, and 18 show the transitions of mask-attentions in policy and state value.

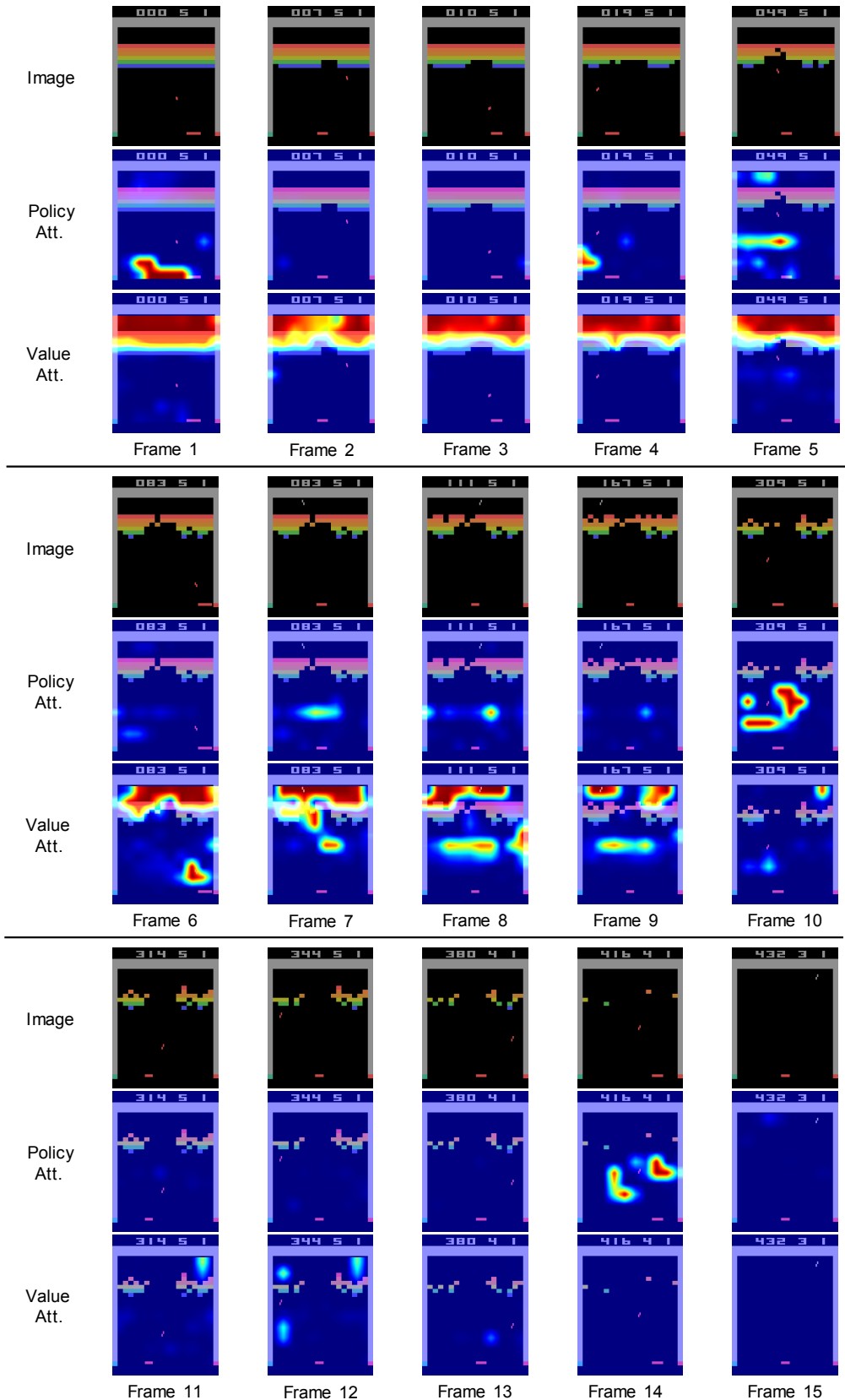

Figure 13: **Transitions of Mask-attentions in policy and value of Breakout**

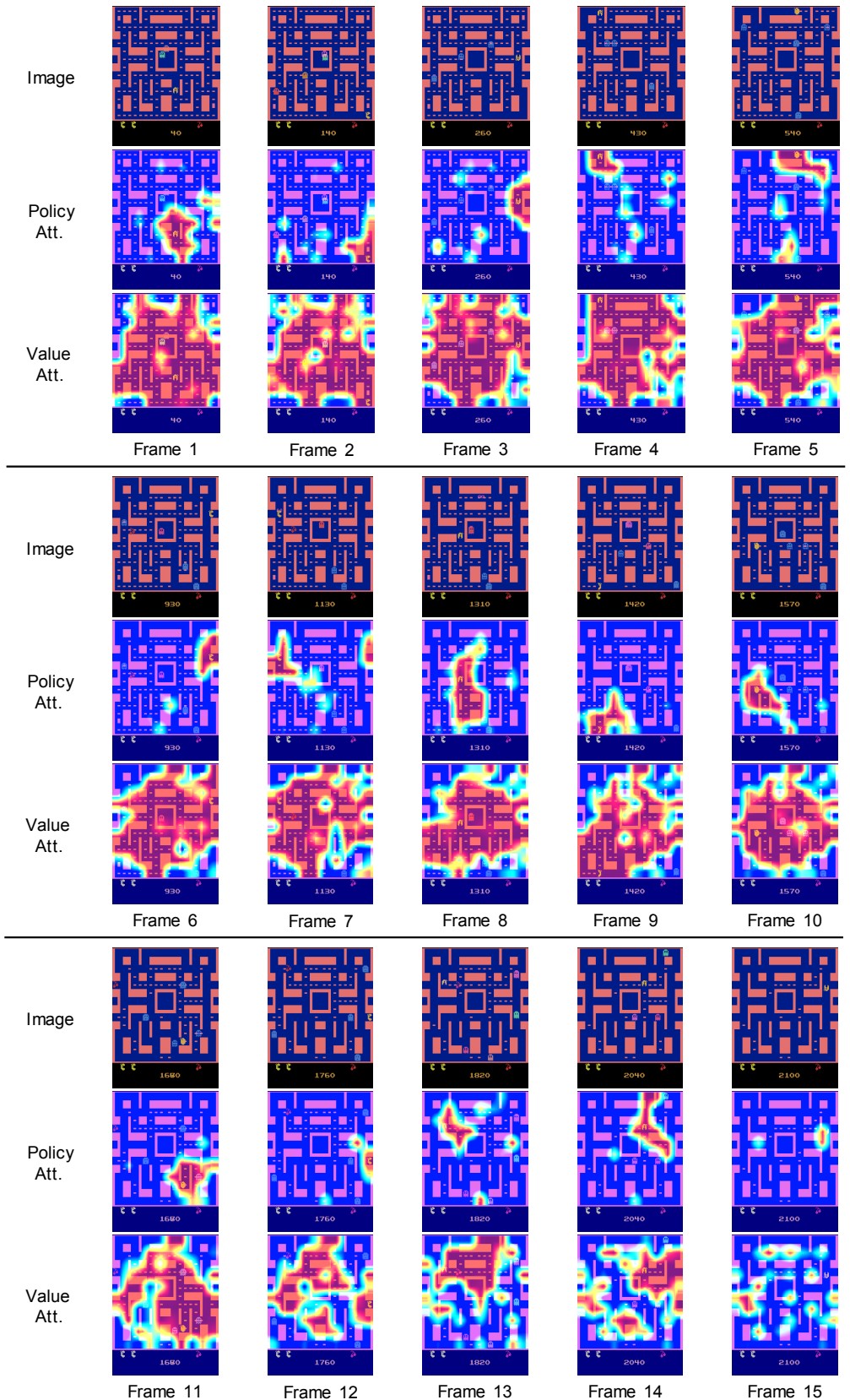

Figure 14: **Transitions of Mask-attentions in policy and value of Ms.Pac-Man**

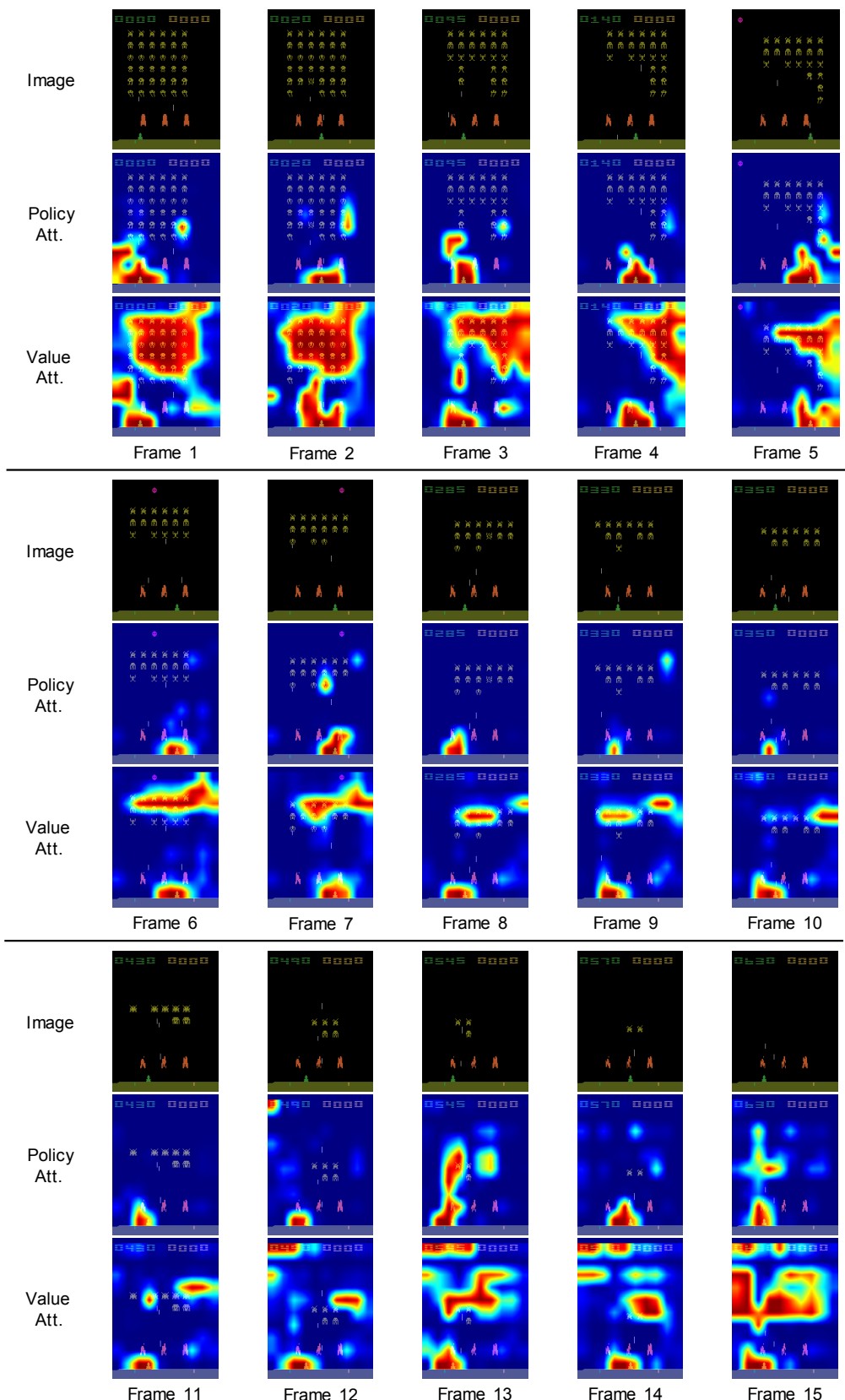

Figure 15: **Transitions of Mask-attentions in policy and value of Space Invaders**

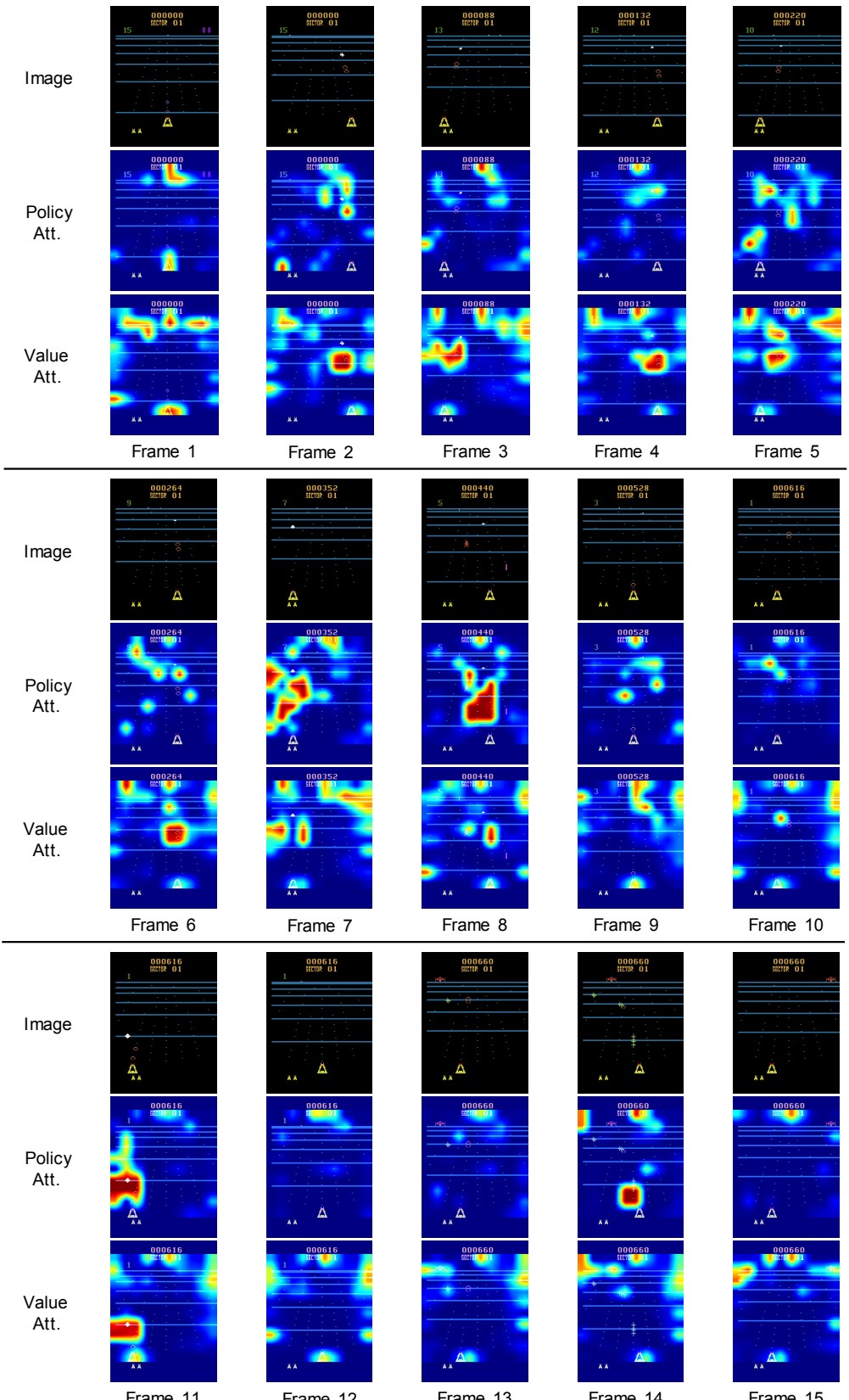

Figure 16: **Transitions of Mask-attentions in policy and value of Beamrider**

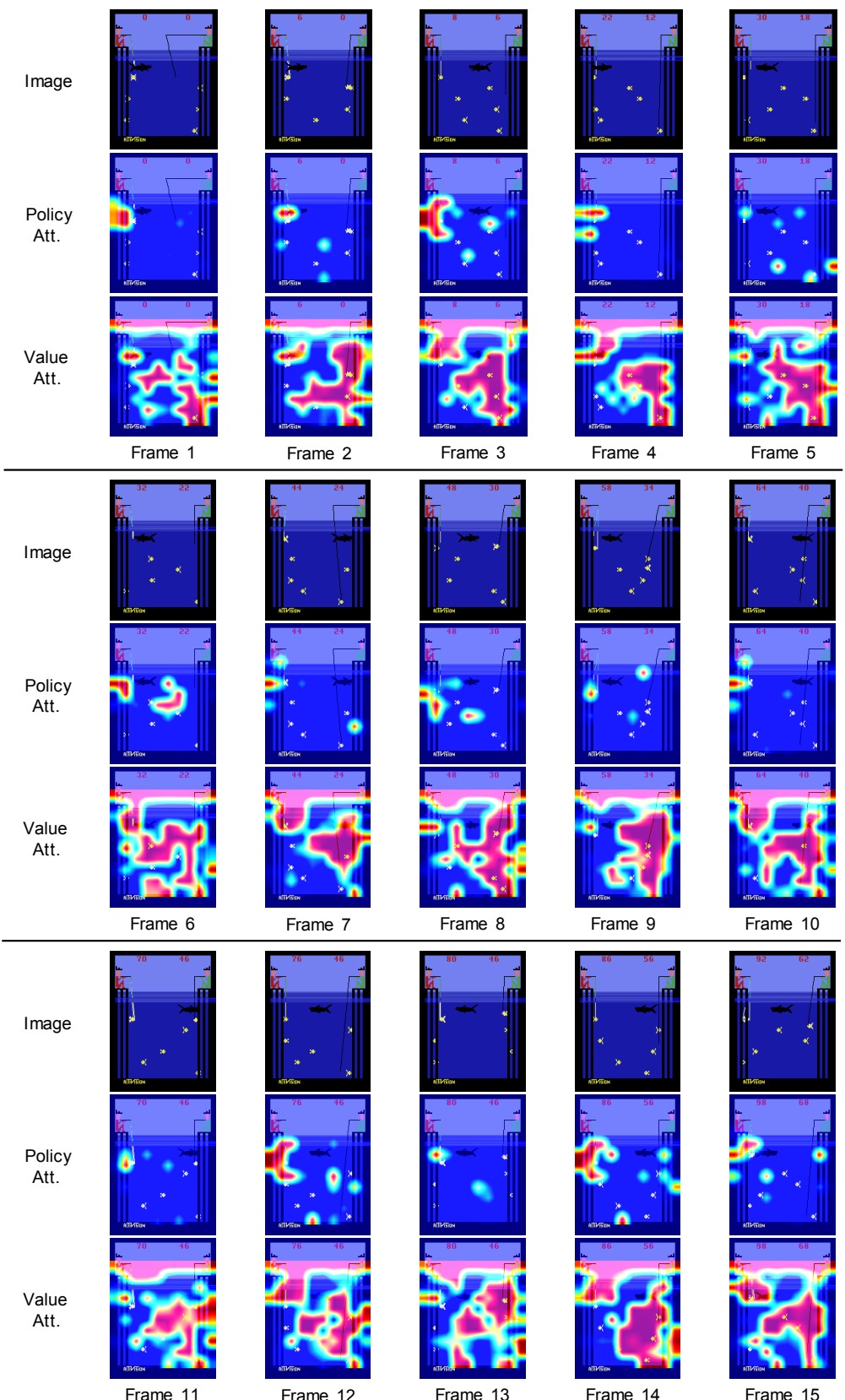

Figure 17: **Transitions of Mask-attentions in policy and value of Fishing Derby**

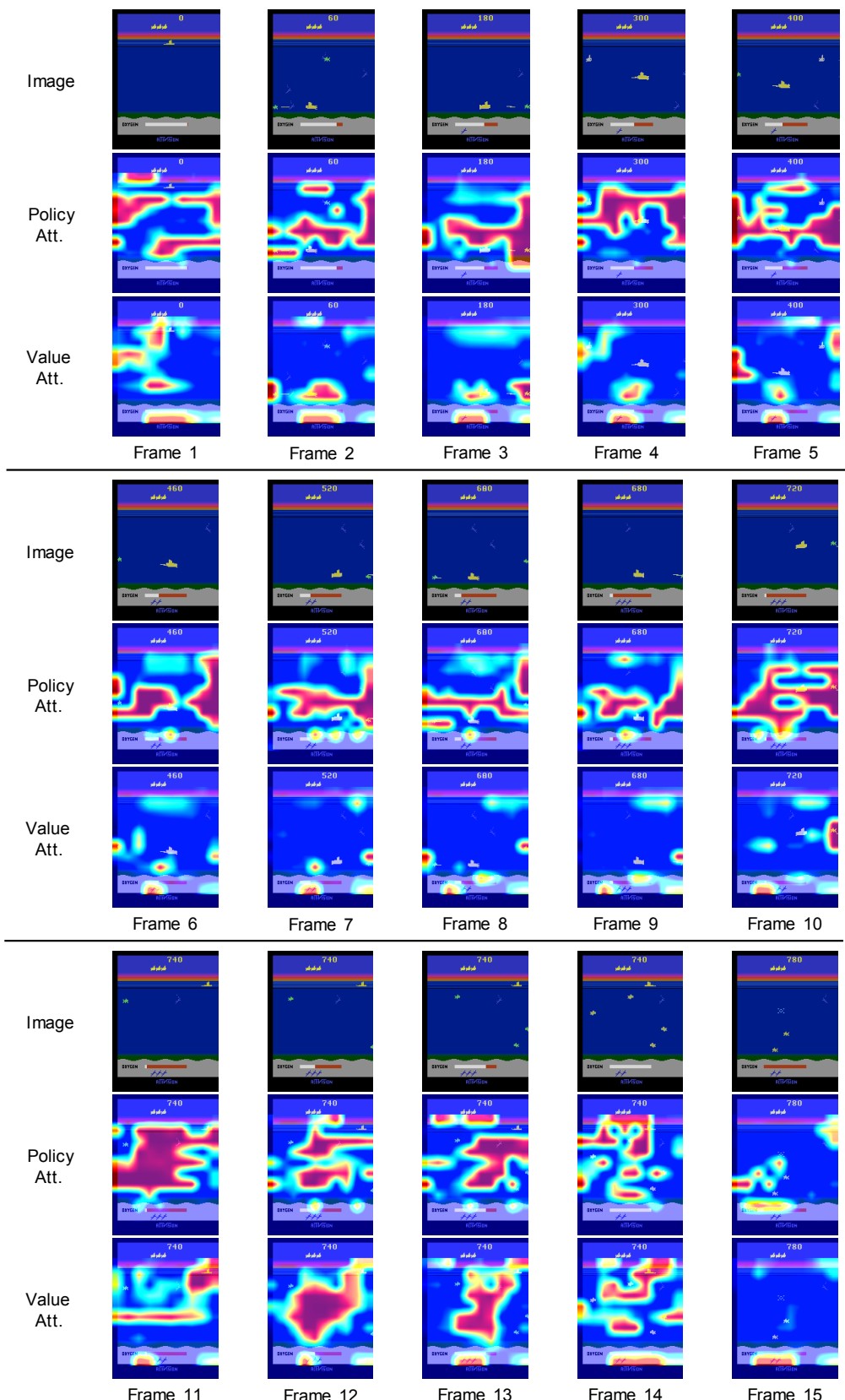

Figure 18: **Transitions of Mask-attentions in policy and value of Seaquest**

**Reaction to new states**

Figures 19 and 20 show the Mask-attention in policy and value for the new state. In this experiment, we confirm the reaction of the attention maps by artificially added new states (i.e., fish and oxygen) that affect the agent's policy and state.

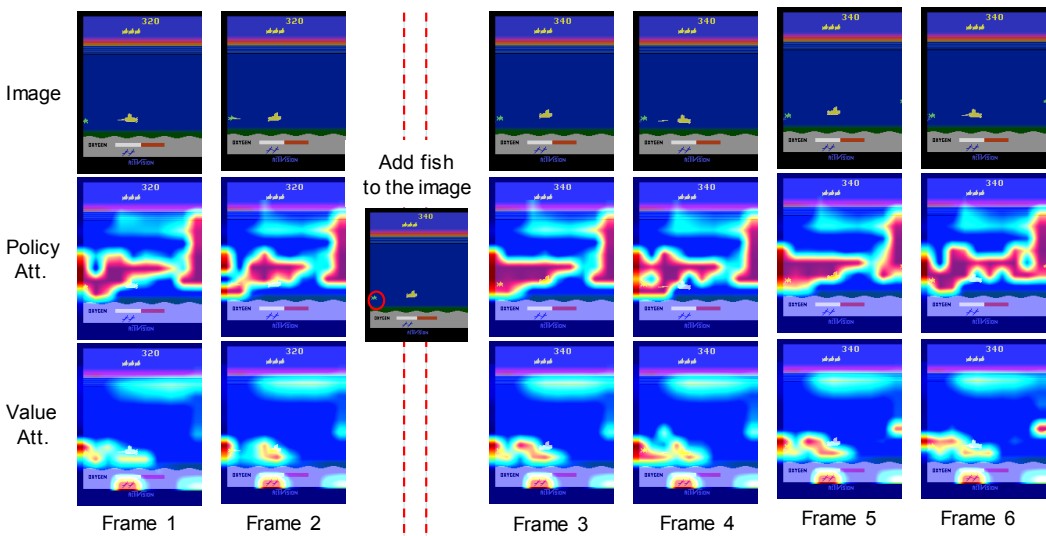

Figure 19: **Reaction to fish-related new states**

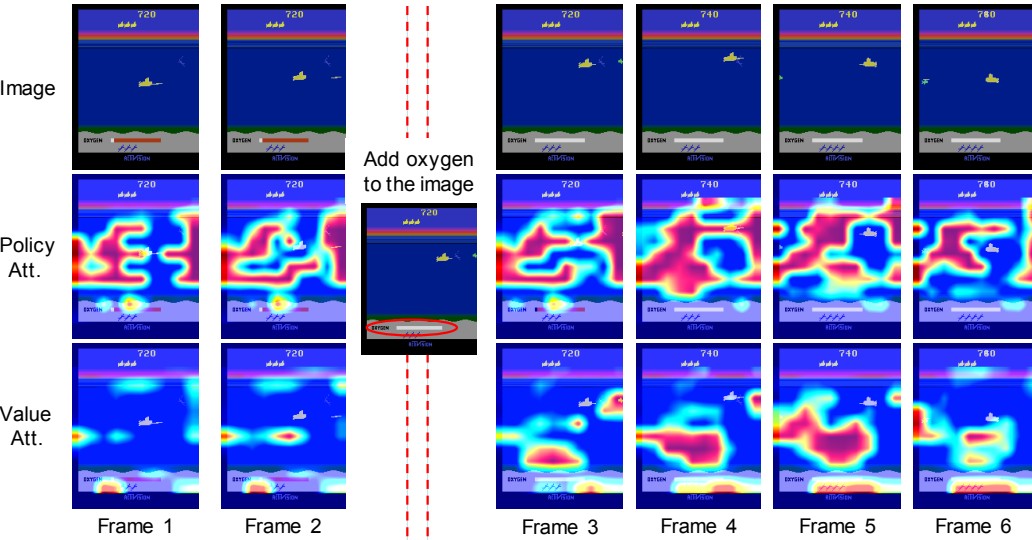

Figure 20: **Reaction to oxygen-related new states**

