# OpenReview forum: "Visual Explanation using Attention Mechanism in Actor-Critic-based Deep Reinforcement Learning"
_ICLR.cc/2021/Conference — Reject_

### Official Review · AnonReviewer4 · 2020-10-28
**Official Blind Review**

**Rating:** 4
**Confidence:** 4

**Review:**

This paper applies the mask attention mechanism on the DRL model (actor and critic), to make the learned policy explainable. The empirical results in Atari 2600 show that the performance of A3C is further improved by implementing mask attention on the actor and critic network separately.

Strength:
- The paper is well-written and easy-to-follow.
- The experiments demonstrate the effectiveness of the attention mechanism in three Atari environments.
- The comparison of the score by the inverting gaze area is interesting.

Weakness:
- The contribution is marginal. The idea is straightforward. It is not a new concept that implementing an attention mechanism in a deep neural network to make it explainable. Applying attention mechanisms to augment the RL agent is also studied in previous work[1, 2, 3]. This key idea of this paper is similar to them, inserting the attention mechanism into the network to explain the RL policy in a top-down fashion. The difference is in the implementation details of the mask-A3C, e.g. using ConvLSTM in the state encoder to keep the spatial information and applying different attention mechanisms for the actor and critic. It is necessary to further discuss and compare these works in the paper.
- The method is only evaluated in three environments. It is necessary to validate it in various environments, especially showing the results in 3D environments (visual navigation, robot arm manipulation). It would be nice to report and discuss the failure cases in the experiments, instead of only the success cases.

Ref:

[1] Mott, Alexander, et al. "Towards interpretable reinforcement learning using attention augmented agents." Advances in Neural Information Processing Systems. 2019.

[2] Manchin, Anthony, Ehsan Abbasnejad, and Anton van den Hengel. "Reinforcement learning with attention that works: A self-supervised approach." International Conference on Neural Information Processing. Springer, Cham, 2019.

[3]Tang, Cheng-Yen, et al. "Implementing action mask in proximal policy optimization (PPO) algorithm." ICT Express (2020).

---

> ### Author Response · Authors · 2020-11-19
> **Thank you for your review.**
>
> Thank you very much for your comments.
>
> The major comments were about the contribution of our work compared with previous works and extensive experiments. The responses to these comments are provided below and we will update our manuscript that reflects these comments.
>
>
> > The contribution is marginal. The idea is straightforward. It is not a new concept that implementing an attention mechanism in a deep neural network to make it explainable. Applying attention mechanisms to augment the RL agent is also studied in previous work.
>
> First, we appreciate to advise the related paper. We have added the description about [A, B] in Section 2.2.
>
> In deep RL, a strategy is an important clue to solve given task and environment. From the viewpoint of the strategy, the state value of actor-critic based model plays a crucial role because the state value is the expected value from current to future states and affects for the future action selections. However, the existing methods outputs attention maps with respect to the instantaneous action selection. Our method based on an actor-critic outputs two attention maps from both policy and value branches. Considering both attention maps, we can understand the basis of an agent's decision-making in more detail.
>
> > The method is only evaluated in three environments. It is necessary to validate it in various environments, especially showing the results in 3D environments (visual navigation, robot arm manipulation). It would be nice to report and discuss the failure cases in the experiments, instead of only the success cases.
>
> We have added additional experiments on another environments (BeamRider, Fishing Derby, Seaquest). The detailed results can be seen in the revised manuscript.
>
> References
>
> [A] A. Mott, et al., “Towards Interpretable Reinforcement Learning Using Attention Augmented Agents”, NeurIPS, 2019.
>
> [B] A. Manchin, et al., "Reinforcement learning with attention that works: A self-supervised approach", ICONIP, 2019.

---

### Official Review · AnonReviewer1 · 2020-10-28
**lacks a clear justification of novelty and contribution**

**Rating:** 5
**Confidence:** 4

**Review:**

This article proposes to include an attention mechanism to Deep RL (focusing on actor-critic architectures), to provide a visual "explanation" of the learned policy. There is solid evidence that attention is an important aspect in perception and learning, and the use of soft attention to improving deep neural network performance has been successful for a variety of tasks such as visual recognition (Wang et al, CVPR'2017) and image captioning - the proposed approach is similar to those.

The proposed approach is to learn attention maps to inhibit part of the visual feature, separately for the value and policy networks. The attention maps being differentiable can be learnt jointly with the rest of the network. The approach is straightforward and similar to attention maps used in, eg, image captioning.

The issue of including attention in RL is not completely unresearched. In particular, it would be valuable to discuss the 2019 DeepMind paper at NeurIPS paper by Mott et al ("Towards Interpretable Reinforcement Learning Using
Attention Augmented Agents") as the claims and purported aims are similar.

The discussion of the type of attention that could be applied (eg, map vs spotlight, soft vs hard) is missing in the article as only one model of attention is evaluated.

The article is generally clear, although some design choices could have been discussed in more details and some arguments are unclear. For example, I did not understand the author's argument of why using bottom-up saliency requires backpropagation.
More importantly, the novelty of the article is not clearly argued: The use of attention maps to analyse and explain deep neural networks is not new in itself, and learning attention maps to improve vision tasks is not new either.

Another issue is that I think the use of the term "explanation" is a bit misleading in this article: the proposed approach provides activation maps, which offer some hints at the system's process, but still require a large amount of human interpretation for an actual explanation.

In sum, the article is fairly well written, but the contribution should be outlined more clearly, and more experimental work could be provided to justify what type of attention is most effective - the relation to some previous works (in particular Mott et al) would also be desirable.

Refs.
Kelvin Xu, Jimmy Ba, Ryan Kiros, Kyunghyun Cho, Aaron Courville, Ruslan Salakhudinov, Rich Zemel, and Yoshua Bengio. Show, attend and tell: Neural image caption generation with visual attention. ICML'2015.

Fei Wang, Mengqing Jiang, Chen Qian, Shuo Yang, Cheng Li, Honggang Zhang, XiaogangWang, and Xiaoou Tang.  Residual attention network for image classification.  CVPR'2017.

Mott, Alexander and Zoran, Daniel and Chrzanowski, Mike and Wierstra, Daan and Jimenez Rezende, Danilo. Towards Interpretable Reinforcement Learning Using Attention Augmented Agents. NeurIPS'2019

---

> ### Author Response · Authors · 2020-11-19
> **Thank you for your review.**
>
> Thank you very much for your comments.
>
> The major commments were about the discussion with the work by Mott et al., the novelty of this paper, the definition of the term “explanation”, and the detailed description of bottom-up saliency with back-propagation. The responses to these comments are provided below and we will update our manuscript that reflects these comments.
>
> > The issue of including attention in RL is not completely unresearched. In particular, it would be valuable to discuss [A] as the claims and purported aims are similar. The discussion of the type of attention that could be applied (eg, map vs spotlight, soft vs hard) is missing in the article as only one model of attention is evaluated.
>
> We appreciate to advise the related paper. Mott et al. [A] also focused on the attention of an actor-critic model. However, their method compute attention maps before policy and value branches. In contrast, our method provides on visual explanations related to policy and attention values as attention maps, respectively. We have added the description about [A] in Section 2.2.
>
> > More importantly, the novelty of the article is not clearly argued: The use of attention maps to analyse and explain deep neural networks is not new in itself, and learning attention maps to improve vision tasks is not new either.
>
> Our contribution relies on that we focused on the policy and value output in a actor-critic method. Analysis and training of DNNs with an attention map have been studied and deep RL models introducing attention have also been proposed. These attention-based deep RL models outputs an attention map from a network model. As we mentioned above, in fact, [A] also output a single attention even though they used actor-critic-based model. Meanwhile, our method provides attention maps with respect to both policy and state values. Therefore, the proposed improve the explainability of deep RL model.
>
> > I think the use of the term "explanation" is a bit misleading in this article: the proposed approach provides activation maps, which offer some hints at the system's process, but still require a large amount of human interpretation for an actual explanation.
>
> In this paper, we use the term “visual explanation” as with some existing works (e.g., [B]). “Visual explanation” visualize where a deep RL model focuses on as attention maps and use them for analyzing the decision-making of the agent.
>
> > I did not understand the author's argument of why using bottom-up saliency requires back-propagation.
>
> We are sorry for the insufficient description. Obviously, not every bottom-up saliencies require back-propagation. The studies [B, C] generate bottom-up saliencies using gradient information computed from network output and target value to train a network. Therefore, those approaches requires back-propagation computation. Meanwhile, [D] is the bottom-up saliency without backprop. that computes the saliency from feature maps obtained from convolutional layer. We will update the part of bottom-up saliency clearly. We have corrected the desceriptions about bottom-up and top-down approaches in Section 1.
>
> References
> [A] A. Mott, et al, “Towards Interpretable Reinforcement Learning Using Attention Augmented Agents”, NeurIPS, 2019.
> [B] R. R. Selvaraju, et al., “Grad-cam: Visual explanations from deep networks via gradient-based localization”, ICCV, 2017.
> [C] G. Samuel, et al., “Visualizing and understanding Atari agents”, ICML, 2018.
> [D] S. Daniel, et al., “Smoothgrad: removing noise by adding noise”, arXiv, 2017.

---

### Official Review · AnonReviewer3 · 2020-10-29
**Nice case studies but need more work**

**Rating:** 5
**Confidence:** 4

**Review:**

Summary:

The paper introduces an attention mechanism into A3C-based reinforcement learning agents to identify the attended visual regions for vision-based reinforcement learning tasks. Specifically, they applied a mask-attention mechanism for the policy and value prediction neural network columns of the A3C model based on the convolutional LSTM (Xingjian et al., 2015). They evaluated the proposed attention mechanism on three selected ATARI games (namely, Breakout, SpaceInvader, and Ms.Pacman) and show that the attention mechanism generates intuitive visual attention in decision-making. They also compared the attention's impact on game scores against some ablative attention mechanism variants.


Pros and Cons:

++ The paper contributes to transparent decision making of reinforcement learning methods by figuring out attended regions in the observational space (pixels). The identified attention regions are intuitive and action-conditional. The cases discussed in the three selected ATARI games are informative.

-- The analysis of the learned attention masks seems selective. Some automatic metrics or systematic studies of different game categories (shooting, maze-like, and ball-and-paddle) may shed light on the learned attention's general property. Current analyses on very sparse time indexes of three selected ATARI games may not provide sufficient evidence or insights to support claims.  Some additional experimental studies on other games or similar domains with high-dimension perceptions would strengthen the paper's contributions.

-- The paper briefly mentions other attention mechanisms for reinforcement learning methods. It seems that some in-depth discussions on the relationship between the proposed approach and the prior art are needed. How is the proposed attention mechanism different from previous ones? Does it address some limitations of previous methods, such as capturing more action-conditional information or more robust to initialization conditions? An additional empirical comparison would also be informative.

-- The motivation and goal on the inverting gaze area are less clear. It would be of interest to see if the attention mechanism would make the learned policy robust to interventions in un-attended regions.

-- The paper has some confusing details. The comparison method named Mask-Attention A3C Double seems identical to the proposed Mask-Attention A3C method. Some clarification on this would be helpful. The paper also has some typos. ``is indicates'', ``our method also learn'', ``is calculates'', etc.

---

> ### Author Response · Authors · 2020-11-19
> **Thank you for your review.**
>
> Thank you very much for your comments.
>
> The cons pointed out were about the extensive experiments to show the strength of the proposed method, the relationship to conventional methods, the detailed motivation of inverting gaze area experiment, and the grammatical problems including the name of the proposed method. The responses to these comments are provided below and we will update our manuscript that reflects these comments.
>
>
> > The analysis of the learned attention masks seems selective.  Some additional experimental studies on other games or similar domains with high-dimension perceptions would strengthen the paper's contributions.
>
> We added more experimental results on another task (BeamRider, Fishing Derby, and Seaquest). Please see the revised manuscript for more details.
>
> > The paper briefly mentions other attention mechanisms for reinforcement learning methods. It seems that some in-depth discussions on the relationship between the proposed approach and the prior art are needed.
>
> We added more detailed discussion about the relationship between the proposed method and conventional methods in the last paragraph of Section 2.2.
>
> > The motivation and goal on the inverting gaze area are less clear. It would be of interest to see if the attention mechanism would make the learned policy robust to interventions in un-attended regions.
>
> In case that the obtained game score by inverting mask-attention does not change, the mask-attention does not contribute for acquired agent's action. In contrast, if the game score significantly decreases, the mask-attention largely contributes for the actions acquiring the game score. Therefore, we conducted the inverting gaze area experiments. For the ease of understand, we added the detailed description in Section 4.4.
>
> > The paper has some confusing details. The comparison method named Mask-Attention A3C Double seems identical to the proposed Mask-Attention A3C method. Some clarification on this would be helpful. The paper also has some typos. is indicates'', our method also learn'', ``is calculates'', etc.
>
> We are sorry for the ambiguous method names and grammatical mistakes. As you pointed out, the “Mask-Attention A3C Double” and “Mask-Attention A3C” are the same, which definitely leads misunderstanding for readers. Therefore, we will remove the term “double” in the revised manuscript. Also, we will correct another grammatical mistakes and typos in the revised paper.

---

### Official Review · AnonReviewer5 · 2020-11-03
**Official Blind Review #5**

**Rating:** 4
**Confidence:** 4

**Review:**

- Summary
    - This paper proposes an interpretable RL agent architecture that uses attention masks to produce visual explanations of the action selected by the policy and output of the value function
    - The authors demonstrate their method on 3 Atari games and use A3C as the training algorithm
- Strengths
    - To the best of the reviewers knowledge, this is the first work to apply this type of visual explanation to RL
    - The interpretable agent performs on par with the black box one.
- Weaknesses
    - How were the points/frames in figure 2 chosen?
    - To my untrained eye, the attention masks in figure 2 aren't very interpretable.  Human studies to verify that the explains actually help the humans understand (or predict) the agent's decision would be very helpful in this regard.
    - I am uncertain that the contribution is enough to warrant publication at ICLR.  While this is the first work I am aware of to apply this type of visual explanation to RL, using attention masks is well known in the literature (Mascharka et al, 2018; Fukui et al, 2019) and it doesn't appear like any considerable modification necessary to apply it to this domain.
    - Using the attention masks to to interpret the decision of the agent based on just the current frame is misleading.  This is because the attention is conditioned on s_t, not o_t, where s_t the output of ConvLSTM(o_t, s_{t-1}).  The consequence is that we do not know whether attention is high for a given location because of the visual information in o_t or the visual information in any other observation. While it is entirely plausible that the most influential location in the ConvLSTM output is most correlated with the current frame, this hasn't been shown.
- Suggestions
    - Show both the frame and the frame with attention in Figure 2.  Currently it can be hard see the content of the frame.
- Overall
    - Overall, I am not convinced the contribution is enough for publication at ICRL.  More importantly, without additional verification the attention masks cannot be used to explain the decision based on the current frame as they are conditioned on the current frame __and__ all previous frames.
- References
    - Mascharka et al, 2018: Transparency by Design: Closing the Gap Between Performance and Interpretability in Visual Reasoning
    - Fukui et al, 2019: Attention Branch Network: Learning of Attention Mechanism for Visual Explanation


## Post Rebuttal

I have increased my rating slightly but still don't think the paper is ready for publication.  I am still not convinced by the quality of the provided visual explanations nor am I convinced that the attention is well correlated with the current frame (the additional experiments provided do help somewhat in this regard, but are not extensive and reasonably inconclusive).

---

> ### Author Response · Authors · 2020-11-19
> **Thank you for your review.**
>
> Thank you very much for your comments.
>
> The major comments were about the criterion of frame and point selection in Figure 2, the contribution of our study, the interpretability of current input frame and attention mask, and distinct figure creation. The responses to these comments are provided below and we will update our manuscript that reflects these comments.
>
>
> > How were the points/frames in figure 2 chosen?
>
> We selected frames from the scene that an agent’s action highly affects with the environment, e.g., the scene where the agent sends a ball back in Breakout. And the neighboring frames are selected as frames 1 and 3. For points, we selected points considering state values from the opening and ending of each game stage. The reason is that the state value is the expected value from current to future states, which contributes the task scene. (Note that the mask-attention in state value is moved to Figure 3 in the revised manuscript.)
>
> > I am uncertain that the contribution is enough to warrant publication at ICLR. While this is the first work I am aware of to apply this type of visual explanation to RL, using attention masks is well known in the literature and it doesn't appear like any considerable modification necessary to apply it to this domain.
>
> In deep RL, a strategy is an important clue to solve given task and environment. From the viewpoint of the strategy, the state value of actor-critic based model plays a crucial role because the state value is the expected value from current to future states and affects for the future action selections. However, the existing methods outputs attention maps with respect to the instantaneous action selection. Our method based on an actor-critic outputs two attention maps from both policy and value branches. Considering both attention maps, we can understand the basis of an agent's decision-making in more detail.
>
> > Using the attention masks to to interpret the decision of the agent based on just the current frame is misleading.
>
> The attention map of our method is output for current network output and not for current frame.
> The most deep RL modes consider current and past states because considering past state achieve accurate and stable actions. For example, deep Q-network-based approaches input several frames into a network and deep RL methods introducing recurrent layer have been proposed including our method. For example, in case that our method output attention map focusing on the past agent location, we can understand the past state is crucial for current decision-making. Therefore, it is not always necessary to correspond the interpretation of an attention map with the current frame.
>
> > Show both the frame and the frame with attention in Figure 2. Currently it can be hard see the content of the frame.
>
> We are sorry for the undistinct figures. We revise figures to show the original input frame without attention mask.

---

> > ### Comment · AnonReviewer5 · 2020-11-19
> > **Clarification on Attention Map Concerned**
> >
> > Thank you for your response.
> >
> > > For example, in case that our method output attention map focusing on the past agent location, we can understand the past state is crucial for current decision-making. Therefore, it is not always necessary to correspond the interpretation of an attention map with the current frame.
> >
> > Can you please elaborate on this and/or point me to the part of the paper that explains this?  As far as I can tell, the attention map is directly overlaid on the current frame to produce the visual explanation.

---

> > > ### Author Response · Authors · 2020-11-20
> > > **Response to AnonReviewer5**
> > >
> > > Thank you very much for your response.
> > >
> > > We show the transitions of attention maps and the corresponding policy and value attention maps in Figs. 13~18.
> > >
> > > From the attention maps for policy, we see that attention occurs on the current location of the agent, or objects (ex, ball, fish, and enemy) that will be interacted with the agent by its movement. Consequently, the transition of the attention maps for the policy through time is changing significantly for each frame. We can say that the attention map for the policy reflects on the current state.
> > >
> > > From the attention maps for value, we see that attention occurs where scoring sources are. For example with the Breakout, Ms.Pac-Man, and Space invader, attention occurs a whole area that covers many scoring objects at the beginning of each game and then changes gradually. Also, we see on seaquest that attention occurs in the area of oxygen gauge on the bottom, and tracked the gauge along with game progress. Therefore, we can say that the attention map for the value reflects on the past and current states.

---

> > > > ### Comment · AnonReviewer5 · 2020-11-20
> > > > **Attention Maps**
> > > >
> > > > I agree that your interpretation feels like it should be correct.  However, it isn't the only possible one.  Since the attention map is conditioned on the output of the ConvLSTM and it is used to attend to a feature map conditioned on the output of the ConvLSTM,  one alternative is that the current frame is not the reason for attention shifting, but the ConvLSTM update.  In the current experiments, I do not see anything that would rule out this possibility.  As an example, "Towards Interpretable Reinforcement Learning Using Attention Augmented Agents" (as pointed out by R1,4) used supporting salience experiments.
> > > >
> > > > For some additional pointers on this topic, this is also similar to the debate in the NLP literature over whether or not attention is explanation, see Jain and Wallace (2019), and Wiegreffe and Pinter (2019).
> > > >
> > > >
> > > >
> > > > References:
> > > >  Jain and Wallace, Attention is not Explanation, NAACL 2019
> > > > Wiegreffe and Pinter, Attention is not not Explanation, EMNLP 2019

---

> > > > > ### Author Response · Authors · 2020-11-24
> > > > > **Response to AnonReviewer5**
> > > > >
> > > > > Thank you for your insightful comments.
> > > > >
> > > > > We performed the same experiment using Seaquest as Mott et al.. We confirmed the influence of RNN on the attention map by visualizing the attention map for new states that do not exist during training.
> > > > >
> > > > > Figures 19 and 20 show the changes in the attention map (policy and state value) produced by the environment with an additional object (e.g. fish, oxygen gauge) injected into the image. Figure 19 shows the transition of the attention map before and after adding fish artificially. From Figure 19, we can see from attention maps of policy that attention occurs the added fish in all frames. The agent attacks the fish and then faces the opposite direction to the fish. Then, the agent attacks the fish again by observing that the fish was not defeated. This is similar to the results of Mott et al. and shows that the agent responds to the current state and behaves appropriately.
> > > > >
> > > > > On the other hand,  from Figure 18, we see that attention maps of value in Seaquest differs from attention maps of policy, and the attention maps of value focus on the oxygen gauge in addition to the fish and the agent. Figure 20 shows the transition of the attention map in a frame with low oxygen content, where the agent is artificially replaced by an oxygen gauge with full oxygen content before replenishing it. From Figure 20, we can see that the attention on the oxygen gauge is gradually observed from frame 3, where an oxygen gauge is added. We also see that the agent’s behavior rises in frames 1 to 4 and falls in frames 5 and 6. Unlike attention maps of policy,  attention map of value reflected not immediately after the addition of the oxygen gauge, but a few frames later. Thus, we can say that state value is considered to be an attention map that strongly reflects the time-series information by ConvLSTM.

---

### Decision · Program_Chairs · 2021-01-07
**Final Decision**

**Decision:**

Reject

**Comment:**

The paper proposes a method to generate attention masks to interpret the performance of RL agents. Results are presented on a few ATARI games. Reviewers unanimously vote for rejecting the papers. R1, R3 give a score of 5, whereas R4, R5 give a score of 4. Their concerns are best explained in their own words:

R1 says, "The use of attention maps to analyze and explain deep neural networks is not new in itself, and learning attention maps to improve vision tasks is not new either."

R3 says, "the analysis of the learned attention masks seems selective. Some automatic metrics or systematic studies of different game categories (shooting, maze-like, and ball-and-paddle) may shed light on the learned attention's general property."

R5 says, "I am still not convinced by the quality of the provided visual explanations nor am I convinced that the attention is well correlated with the current frame (the additional experiments provided do help somewhat in this regard, but are not extensive and reasonably inconclusive"

In their rebuttal, to address R1's concern authors suggested that the use of attention on both value and policy networks is novel. This is not sufficient, because it does not show why such attention maps are more useful than ones proposed by prior work. As suggested by reviewers, a systematic study or a human study clearly showing that the proposed method adds more interpretability is critical. However, this is missing.  In response to R3, the authors provided experiments on more games. But this is not the point -- because it's not about the number of environments in which experiments are provided, but rather the nature of the analysis that is performed. Finally, R5 comments that it's unclear whether attention actually provided interpretability or not.

Due to the lack of convincing analysis that demonstrates the utility of the proposed method in advancing the understanding of decisions made by RL agents, I recommend that the paper be rejected.